# The Role of Patient-Derived Organoids in Triple-Negative Breast Cancer Drug Screening

**DOI:** 10.3390/biomedicines11030773

**Published:** 2023-03-03

**Authors:** Iason Psilopatis, Amalia Mantzari, Kleio Vrettou, Stamatios Theocharis

**Affiliations:** 1Department of Gynecology, Charité—Universitätsmedizin Berlin, Corporate Member of Freie Universität Berlin and Humboldt—Universität zu Berlin, Augustenburger Platz 1, 13353 Berlin, Germany; 2First Department of Pathology, Medical School, National and Kapodistrian University of Athens, 75 Mikras Asias Street, Bld 10, Goudi, 11527 Athens, Greece

**Keywords:** organoid, triple-negative, breast cancer, three-dimensional, cell culture

## Abstract

Triple-negative breast cancer (TNBC) is one of the most aggressive breast cancer subtypes, with a grave prognosis and few effective treatment options. Organoids represent revolutionary three-dimensional cell culture models, derived from stem or differentiated cells and preserving the capacity to differentiate into the cell types of their tissue of origin. The current review aims at studying the potential of patient-derived TNBC organoids for drug sensitivity testing as well as highlighting the advantages of the organoid technology in terms of drug screening. In order to identify relevant studies, a literature review was conducted using the MEDLINE and LIVIVO databases. The search terms “organoid” and “triple-negative breast cancer” were employed, and we were able to identify 25 studies published between 2018 and 2022. The current manuscript represents the first comprehensive review of the literature focusing on the use of patient-derived organoids for drug sensitivity testing in TNBC. Patient-derived organoids are excellent in vitro study models capable of promoting personalized TNBC therapy by reflecting the treatment responses of the corresponding patients and exhibiting high predictive value in the context of patient survival evaluation.

## 1. Introduction

Breast cancer represents the leading malignancy and the second most common cause of cancer death in women in the United States [1]. Triple-negative breast cancer (TNBC) accounts for approximately 15% of all breast cancers and describes a type of breast cancer whose cells do not overexpress the human epidermal growth factor receptor 2 (HER2) or receptors for estrogen and progesterone [2]. Patients with TNBC may present with signs and symptoms similar to those of other common breast cancer types, yet TNBC shows a more aggressive behavior with higher metastasis and recurrence rates [3]. As a result, TNBC is commonly diagnosed in advanced tumor stages, with the 5-year survival rate dropping to 12% in the case of distant metastases [4]. Breast-conserving surgery, followed by postoperative radiotherapy, and total mastectomy represent feasible therapeutic options for localized TNBC in the initial stage [5]. Chemotherapy is the main method of systemic therapy and a key element of combined treatment [6]. Currently applied first-line chemotherapeutic regimens consist of platinum compounds, taxanes, anthracyclines, as well as antimetabolites [7]. The lack of estrogen receptor (ER), progesterone receptor (PR), and HER2 expression renders endocrine and trastuzumab therapy ineffective for TNBC patients, thus necessitating the application of targeted drugs, antibody-drug conjugates, or immunotherapy in advanced stage TNBC [4].

Organoids have piqued the interest of many researchers in oncology as they allow for the three-dimensional reconstruction of (ab-)normal tissues in vitro from pluripotent or tissue-resident stem cells or differentiated normal or cancer cells, thus mimicking the biological and functional profiles of healthy or cancer tissues [8,9]. Following their initial introduction in 2009 as groundbreaking three-dimensional primary tissue culture models, tumor organoids have been derived from tissue or liquid biopsies and resected specimens of various cancers and have been used for human developmental biology, disease modeling, tissue engineering, regenerative and personalized medicine, as well as drug screening [10,11,12]. Their establishment requires scaffolding or scaffold-free techniques in order to forestall direct physical contact with the plastic dish [13]. Matrigel, a heterogeneous and gelatinous protein mixture purified from Engelbreth-Holm-Swarm mouse sarcoma cells that resembles the natural extracellular matrix (ECM), is the most commonly used scaffold [14]. So far, Matrigel or similar hydrogels have been successfully employed for the generation of gastrointestinal, salivary gland, hepatic, pancreatic, brain, retinal, renal, pulmonary, or gynecological organoids [15]. The current review of the literature focuses on the distinct role of breast, and more precisely TNBC, patient-derived organoids.

TNBC is an aggressive form of breast cancer with poor patient outcomes, underlining the unmet clinical need for efficient targeted therapies and better model systems. Even though two-dimensional models have been invaluable, they are not suitable for studying patient-specific tumor biology and drug response. The present work aims at investigating the role of patient-derived TNBC organoids in drug sensitivity testing with a view to exploring novel promising perspectives in terms of TNBC therapy. The literature review was conducted using the MEDLINE and LIVIVO databases. Solely original research articles and scientific abstracts written in the English language that explicitly reported on the development of three-dimensional TNBC patient-derived organoid models were included in the data analysis. Studies incorporating two-dimensional or spheroid TNBC models, as well as studies not clearly stating the use of patient-derived TNBC organoid models, were excluded. The search terms “organoid” and “triple-negative breast cancer” were employed, and we were able to identify a total of 69 articles published between 2013 and 2022, after the exclusion of duplicates. A total of 33 were discarded in the initial selection process after abstract review due to topic irrelevance. The full texts of the remaining 36 publications were evaluated, and after detailed analysis, a total of 25 relevant studies published between 2018 and 2022 that met the inclusion criteria were selected for the literature review. Figure 1 presents an overview of the aforementioned selection process.

## 2. Generation of Breast (Cancer) Organoids

The generation of breast (cancer) organoids is rather complicated and requires high expertise in this field. Given that the concentration of the collagen I fibrils in the mixed gel plays a significant role in the development of a human breast organoid model comprising the breast tissue microenvironment, synthetic scaffolds, including polymers that retain the mechanical properties of the tumors, are much less commonly utilized in breast organoid culture [16]. Specifically, breast organoids have been reported to accurately recapitulate the in vivo breast microenvironment and provide a trustworthy insight into the factors affecting signaling transduction, gene expression, and tissue remodeling, hence facilitating the study of normal mammary gland development and tumorigenesis [17]. Interestingly enough, Mohan et al., discussed significant mammary organoid studies with associated protocol details, and pointed out the similarities and differences observed in matrix type, medium components, plating techniques, as well as layering methods. Additionally, the same study group described the limitations of current breast organoid models, highlighting numerous contributing factors such as differential centrifugation, cell straining, or multiple cell types [17]. Sachs et al., successfully developed a vigorous protocol that allows for the long-term culturing of human mammary epithelial organoids and, by extension, the generation of primary and metastatic breast cancer organoid lines [18]. More precisely, after obtaining patient-derived breast cancer tissue samples, breast cancer cells need to be isolated through a combination of mechanical disruption and enzymatic digestion. These cells are then plated in adherent basement membrane extract drops and overlaid with an adjusted breast cancer organoid culture medium. The addition of the mitogen Neuregulin 1 to the breast cancer organoid medium not only enables the efficient establishment of breast cancer organoids but also facilitates their long-term expansion for multiple passages. Accordingly, the addition of the specific Rho-associated coiled-coil containing protein kinase (ROCK) inhibitor Y-27632 also ameliorates culture conditions by allowing long-term proliferation of tumor epithelial cells in vitro. On the contrary, addition of Wnt-3A does not seem to effectively contribute to culture condition improvement. Elevated epidermal growth factor (EGF) concentrations might augment proliferation but correlate with gradual breast cancer organoid sinking through basement membrane extract and three-dimensional organization loss, whereas high SB202190 concentrations lead to a profound decrease in organoid establishment efficiency. The approaches the authors used to describe fidelity the of the patient-derived organoids ranged from blinded histopathological analysis and karyotyping to whole-genome sequencing (WGS) and RNA-sequencing. The aforementioned protocol paved the way for the creation of a representative collection of well-characterized breast cancer organoids that matched the histopathology, ER, PR, and HER2 status of the tumor of origin, as well as populated the main gene-expression-based classification categories [18]. Similarly, Dekkers et al., described a protocol for the efficient derivation and long-term cultivation of human normal and breast cancer organoids [19]. Both the growth of newly established organoid cultures in appropriate expansion media and the use of R-spondin-1, Noggin, and Wnt-3A-conditioned media seem to represent minimum requirements in the context of breast organoid culture conditions. Basement membrane extract (BME) polymerizes at temperatures above 10 °C and may mimic the extracellular matrix, hence promoting a three-dimensional organoid amplification. To be more accurate, organoid establishment from resections of normal breast or tumor tissue is followed by organoid maintenance or freezing for long-term storage. Lipofectamine-based transfection, electroporation-based transfection, or lentiviral transduction represent three different employed methods that allow for genetic manipulation as well as (clonal) organoid selection. The mean duration for the generation of a breast organoid culture, from tissue isolation until the first passage, may be estimated at one to three weeks, with the best moments for passage (via either single cells or fragments) being the phase before organoid death in the center of the BME drop or an organoid diameter larger than 300 μm. Genetical manipulation of an organoid culture as well as generation of a selected clonal line at passage one will take at least two weeks. Organoid culture expansion for transplantation usually requires more than one month, depending on different factors such as the number of mice injected or the organoid proliferation rate. Notably, this injection-based method for estrogen pellet implantation and breast cancer organoid xenotransplantation may, on the one hand, elude the necessity for advanced and highly complicated surgical procedures but, on the other hand, is relatively laborious and costly in comparison with the majority of other available in vitro breast cultures [19].

## 3. The Role of Patient-Derived Organoids in the Understanding of TNBC Pathogenesis

Since their initial application, patient-derived organoids have provided great insights into the pathophysiology of numerous cancer entities and, especially, shed light on obscure and otherwise unexplainable aspects related to the pathogenesis of scarce or complex tumors [20,21,22,23]. TNBC belongs to the multifaceted and, consequently, aggressive subtypes of breast cancer, as it does not abide by the well-understood hormonal pathways that characterize the more common hormone-receptor positive breast cancer entities but develops and spreads through diverse and extremely complicated molecular mechanisms [24]. In this context, several study groups have successfully generated TNBC patient-derived organoids and attempted to explore the signaling pathways involved in TNBC pathogenesis. Mazzucchelli et al., were among the first study groups to launch a new protocol for the establishment of patient-derived organoids from surgical and biopsy breast cancer samples that also clearly incorporated TNBC specimens. More precisely, patient-derived organoid cultures were obtained from tissue sample resection with a success rate of 87.5%, with one out of the 21 generated patient-derived organoids histologically belonging to TNBC. Sufficient material for patient-derived organoid establishment was obtained from each of the four incorporated TNBC patients that underwent standard core biopsy for lesion characterization prior to neoadjuvant treatment initiation, accordingly [25]. Interestingly, Dekkers et al., reported a relatively low efficiency for TNBC patient-derived organoid establishment in comparison with other breast cancer subtypes, which was attributed to the aggressive and genetically unstable features of TNBC cells that maladapt to in vitro culture conditions [19]. Fang et al., mixed TNBC cells with different Mal, T Cell Differentiation Protein 2 (MAL2) expression levels with cancer-associated fibroblasts from the same tumor tissue to form TNBC patient-derived organoids. After organoid coculture with pre-activated autologous CD8+ T cells, MAL2 depletion was found to enhance CD8+ T cell cytotoxicity [26]. Furthermore, Bhatia et al., developed long-term TNBC patient-derived organoids that imitated the extensively studied and evidently proven features of this aggressive MYC-driven, basal-like breast cancer and were largely comprised of luminal progenitor-like cells exhibiting hyperactivation of NOTCH and MYC signaling [27], while Chew et al., described low or undetectable Fibroblast Growth Factor Receptor 4 (FGFR4) immunohistochemical staining in TNBC patient-derived organoids [28]. Altogether, the study of TNBC patient-derived organoids may pave the way for novel and reliable discoveries in the field of TNBC pathogenesis that other in vitro or even in vivo culture models do not allow. Especially the generation and recapitulation of the tumor (immune) microenvironment, including neoplastic cells and non-cancerous host components, may provide a critical insight into tumor behavior and contribute to a better understanding of TNBC genesis, progression, and metastasis. 

## 4. The Use of TNBC Organoids for Chemotherapy Sensitivity Testing

A great number of studies have, so far, investigated the use of TNBC organoids for chemotherapy sensitivity testing.

Campaner et al., established patient-derived organoids from different breast cancer subtypes and tested the effect of standard therapies on their viability. TNBC organoid viability was reduced after docetaxel application but remained unaffected by tamoxifen treatment [29]. Moreover, Chen et al., derived organoids from patients with TNBC, drug-resistant, or metastatic tumors. Patient-derived organoid pharmaco-phenotyping was found to highly correlate with clinical outcomes and reflect previous treatment responses of the corresponding patients. Importantly, patient-derived organoids may predicate special sensitive drugs for personalized therapy, given that all treatments including at least one drug predicated to be sensitive by patient-derived organoids, achieved partial response, stable disease, or long disease-free survival (DFS), in TNBC patients [30]. Similarly, Shu et al., matched the organoid drug sensitivity data with the patient’s clinical results and outlined the consistency of the clinical response of TNBC patients with the response of tissue-derived organoids to neoadjuvant chemotherapy (docetaxel, epirubicin) [31]. These results outline the applicability of patient-derived TNBC organoids as mighty in vitro study models for the evolution of individualized therapy concepts.

Additionally, Cromwell et al., generated organoids derived from patient-derived xenograft tissue from serial transplantation of metaplastic breast cancer with a TNBC subtype in immunocompromised mice. After the preparation of two passages of the patient-derived xenograft organoids, the flowchip system in combination with high content imaging was used to evaluate the effects of romidepsin, trametinib and paclitaxel. In comparison with respective scalable tumoroids, the results were found to be consistent, including a high IC_50_ paclitaxel value [32]. Analogously, Matossian et al., used a similar organoid model and described paclitaxel’s ability to downregulate CD44 and E-cadherin expression in a dose-dependent manner [33]. These studies combined the benefits of both xenografts and organoids and retained tumor-stroma interactions, which significantly contribute to tumorigenesis.

Liu et al., established TNBC patient-derived organoid cultures and treated them with paclitaxel in both the presence and absence of the Rab effector in vesicle transport, Synaptotagmin-like 4 (SYTL4). Hoechst/PI staining showed that paclitaxel application induced apoptosis in the treated patient-derived organoids, especially after SYTL4 knockdown [34]. Wang et al., utilized TNBC patient-derived organoids to explore the roles of the candidate circular ribonucleic acid (RNA) and showed that patient-derived organoids with higher expression levels of a circular RNA acting as a chemo-resistance inhibitor in TNBC tended to be more sensitive to doxorubicin and to have lower IC50 values [35]. Last but not least, Yu et al., proved the association between protein levels of deoxyribonucleic acid methyltransferase (DNMT) and the response to decitabine treatment in patient-derived xenograft organoids originating from chemotherapy-sensitive and -resistant TNBC [36]. Taken together, these study results underline the advantages of patient-derived organoid cultures for the experimental investigation of unknown aspects in terms of TNBC chemotherapy.

Impressively, patient-derived organoids may also be employed for oncolytic virotherapy testing, thereby potentially endorsing the establishment of this novel type of multi-mechanistic targeted agent for TNBC immunotherapy. With a view to quantifying their susceptibility to oncolytic virotherapies, Behrens et al., employed a sample of the actual original oncolytic Urabe MuV clinical trial virus stock (MuV-U-Japan) and reported that the two isolates, MuV-UA and MuV-UC, exhibited efficient killing activity against TNBC patient-derived xenograft cell lines grown as three-dimensional organoids that were resistant to standard anthracycline- and taxane-based chemotherapy [37]. In the same context, Huang et al., explored the tumor suppressor and immune-activating effects of an in-situ DC vaccine (HELA-Exos) in immunocompetent mice and TNBC patient-derived organoids and showed that HELA-Exos possessed a profound ability to augment type one conventional dendritic cell (cDC1) antigen cross-presentation and tumor-reactive CD8+ T-cell generation, hence leading to potent TNBC inhibition [38]. 

The use of patient-derived organoids for chemotherapy sensitivity testing in TNBC cancer is summarized in Table 1.

## 5. The Use of TNBC Organoids for Alternative and Targeted Therapy Sensitivity Testing

Various studies have examined the use of TNBC organoids for alternative and targeted therapy sensitivity testing, as well.

Conway et al., utilized two patient-derived organoid cultures from primary basal-like TNBC tumors and stated that small molecule inhibitors of the transcription factors Glucocorticoid receptor (GR) and signal transducer and activator of transcription 3 (STAT3) lead to a significant, synergistic, dose-dependent decrease in TNBC cell growth, measured by ATP levels [39]. Furthermore, Ge et al., selected two TNBC patient-derived organoids and suggested that tektin4-deficient organoids favor the efficacy of histone deacetylase 6 (HDAC6) inhibition via the selective ACY1215 inhibitor in TNBC [40]. Additionally, Guillen et al., leveraged matched patient-derived xenografts and patient-derived xenograft organoids for drug screening. 50% of the employed TNBC patient-derived xenograft organoid lines showed remarkable sensitivity to the second mitochondrial-derived activator of caspases (SMAC) mimetic birinapant, whereas the remaining TNBC lines were resistant to high birinapant doses. Moreover, based on the promising anti-cancer effects of microtubule dynamics inhibitors in a TNBC organoid model, a patient with TNBC with early metastatic recurrence received eribulin treatment, resulting in a complete response for the individual as well as longer progression-free survival (PFS) and time to next systemic therapy periods than previous therapies [41]. These results confirm the statement that patient-derived organoids may act as excellent in vitro study platforms for the development of individualized, even alternative or targeted, treatment concepts. Notably, Jung et al., cultured patient-derived organoids from four TNBC patients and found nicotinamide (NAM), a water-soluble amide form of niacin, to inhibit TNBC organoid growth as well as disrupt the three-dimensional spheroid structures [42]. Kurani et al., carried out assays of drug action in two different patient-derived xenograft organoid cultures, and concluded that the disruptor of telomeric silencing-1 like histone lysine methyltransferase (DOT1L) inhibitor EPZ-5676 abrogates clonogenic patient-derived xenograft organoid culture growth [43], while Li et al., generated patient-derived organoid models to assess the therapeutic benefits of targeting *Cyclin Dependent Kinase 16* (*CDK16*) in TNBC, and underlined that *CDK16* knockdown suppressed organoid growth and Ki67 expression [44]. Furthermore, Liu et al., developed organoids from primary TNBC tissues and reported combined treatment of RU.521, a cyclic GMP–AMP synthase (cGAS) inhibitor, and afatinib or gefitinib to suppress organoid growth [45]. After testing the small-molecule antagonist of exportin-1, LFS-1107, in five patient-derived organoids derived from TNBC patients, Liu et al., highlighted that LFS-1107 effectively inhibited patient-derived organoid proliferation [46]. Furthermore, Parsyan et al., observed that radiation therapy combined with Polo-Like Kinase 4 (PLK4) inhibitor CFI-400945 exhibit a synergistic anti-cancer effect in TNBC patient-derived organoids [47]. Saatci et al., generated organoid cultures of a doxorubicin-resistant TNBC tumor and demonstrated that lysyl oxidase (LOX) inhibitors combined with doxorubicin synergistically induce organoid shrinkage. Combination treatment with doxorubicin and β-Aminopropionitrile (BAPN) led to a significant reduction in organoid size in a primary organoid model developed from a treatment-naïve TNBC patient, respectively [48]. As such, patient-derived TNBC organoids seem to provide researchers with great opportunities to also investigate the efficacy of novel agents in combination with well-established chemotherapeutics, hence exploring potential synergistic effects. Moreover, Sudhakaran et al., sought to investigate the effect of the dietary flavone apigenin on TNBC patient-derived xenograft organoids and found that apigenin decreases both their growth and viability at concentrations achievable in vivo [49]. Wu et al., treated four different organoid models with either dimethyl sulfoxide (DMSO) or MS023. Microscopic evaluation and PrestoBlue staining revealed that organoids with higher basal interferon gene expression were more sensitive to type I protein arginine methyltransferase (PRMT) inhibition, whereas quantitative real-time polymerase chain reaction (qRT-PCR) confirmed the upregulation of T1-helper cell chemokines and antigen presentation genes [50]. After performing experiments on patient-derived organoids, Xiao et al., indicated that the sphingosine kinase 1 (SPHK1) inhibitors PF-543 and fingolimod showed significantly higher efficacy in luminal androgen receptor (LAR) positive TNBC [51]. Additionally, Yang et al., employed patient-derived organoids, and validated the hypothesis that TNBC tumors with macrophage receptor with collagenous structure–TST (MARCO-TST) expression are more sensitive to bromodomain and extra-terminal protein inhibitors [52], whereas Zhang et al., developed patient-derived organoid models from two TNBC patients and concluded that pharmacologic inhibition by the covalent CDK14 inhibitor FMF-04-159-2 results in both organoid size and number reduction, decreased cell proliferation, and apoptosis induction [53]. Altogether, these observations reveal the countless advantages that patient-derived TNBC organoids might offer as suitable in vitro study platforms for alternative and targeted therapy sensitivity testing.

The use of patient-derived organoids for alternative and targeted therapy sensitivity testing in TNBC is summarized in Table 2.

## 6. Conclusions

Breast organoids undoubtedly represent the most useful in vitro tools for the understanding of not only the normal human mammary gland but also breast cancer development [17,54]. To date, numerous protocols have been established for the generation of such three-dimensional breast organoid cultures [17]. Their main contributions to breast cancer research have already been summarized in past comprehensive review articles that outline the role of these exceptional models in capturing breast cancer-associated molecular pathways, investigating the relevant pathophysiology, and predicting drug response to address treatment efficacy in breast cancer patients [55,56,57]. Nonetheless, no review article has, so far, been published on the particular role of organoids in TNBC research, even though this aggressive breast cancer subtype may still be characterized as a “black box” with diverse unknown aspects that render its therapeutic approach challenging. 

The present work, for the first time, focuses on the advantages of particularly patient-derived organoids in the context of TNBC drug sensitivity testing, given that, in comparison with cancer cell lines, xenograft models, or two-dimensional culture models, they succeed in scholastically recapitulating basic features of the primary tumor and its microenvironment, including epithelial-mesenchymal transition (EMT) [58]. As screening platforms for standard chemotherapy drug agents, patient-derived organoids offer the opportunity to test the efficacy of standard, widely applied, Food and Drug Administration (FDA)-approved chemotherapeutics, hence allowing for a trustworthy (re-)evaluation of each agent and a realistic selection of TNBC patients who may profit from each therapeutic regime. Importantly, resistance to certain chemotherapeutics may be pretested under in vitro conditions, thus avoiding ineffective chemotherapy application to unsuitable patients and reducing undesirable side effects. Additionally, combinational therapy with novel experimental agents may be tested out in order to develop groundbreaking regimes that would redefine TNBC treatment by bypassing acquired resistance or enhancing tumor cell sensitivity to certain chemotherapeutics. The use of patient-derived organoids for alternative and targeted therapy sensitivity testing may revolutionize TNBC therapy, respectively. Patient-derived organoids have given rise to an excellent drug platform that may be employed to study the mechanism of action and potential of agents other than standard chemotherapy for TNBC, examine their efficacy in certain chemotherapy-resistant TNBC subtypes, as well as compare them with first-line therapeutic approaches. Figure 2 compactly depicts the current FDA-approved treatment agents for TNBC [59] versus the drugs tested in TNBC patient-derived organoids and summarized in the present work. 

In spite of the great potential, the generation of TNBC patient-derived organoids is associated with certain drawbacks that hinder their widespread establishment as the principal preclinical model for TNBC research and drug sensitivity screening. Firstly, TNBC patient-derived organoid development has technical limitations, as the required preparation is both time-consuming and expensive, in comparison with other in vitro, or even in vivo, currently available culture models. Secondly, standard protocols for TNBC patient-derived organoids have yet to be determined and widely established, given that the currently available protocols for breast (cancer) organoids may not be applied to each and every breast cancer subtype, including the aggressive and complicated TNBC. Thirdly, the reproduction of the convoluted TNBC tumor microenvironment might be challenging, thereby partially impairing the functionality and heterogeneity of the organoid model as well as altering the patient-derived organoid response to drug screening assays. Of note, these disadvantages are not limited to only TNBC patient-derived organoids. Similar limitations have been described for other cancer entities as well, thus outlining the imperative need for further research in the field of organoid culture optimization [22,23].

Altogether, patient-derived organoids could promote personalized TNBC treatment, as individual patient-derived organoids reflect the treatment responses of the corresponding patients and have a high predictive value in terms of patient survival evaluation. Despite the numerous advantages, patient-derived organoids still represent in vitro study models. Therefore, the drug screening results should be compared with the clinical outcomes in a standardized manner and examined in large, adequately designed randomized clinical trials, with a view to assessing the organoids’ capacity to predict the efficacy of chemotherapy, alternative therapy, and targeted therapy in the clinical setting.

## Figures and Tables

**Figure 1 biomedicines-11-00773-f001:**
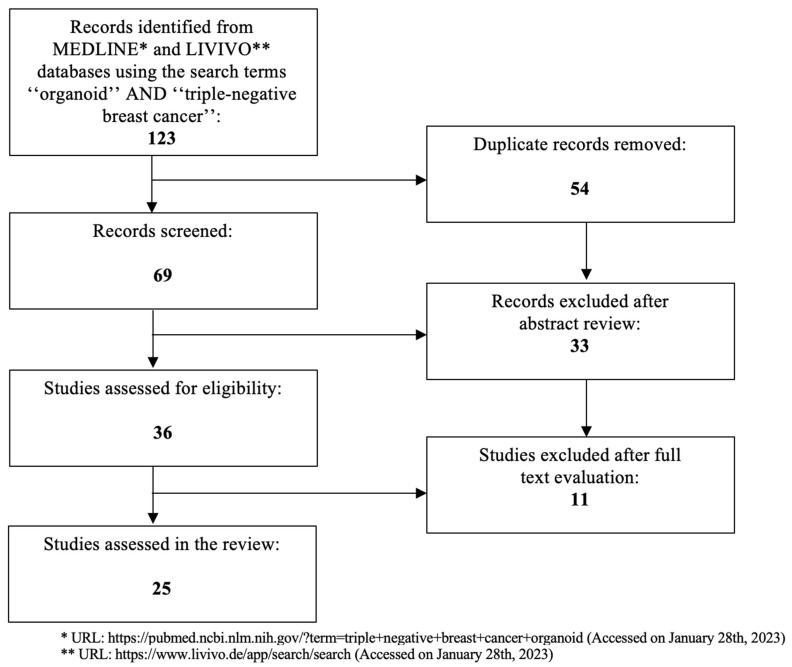
PRISMA flow diagram visually summarizing the screening process.

**Figure 2 biomedicines-11-00773-f002:**
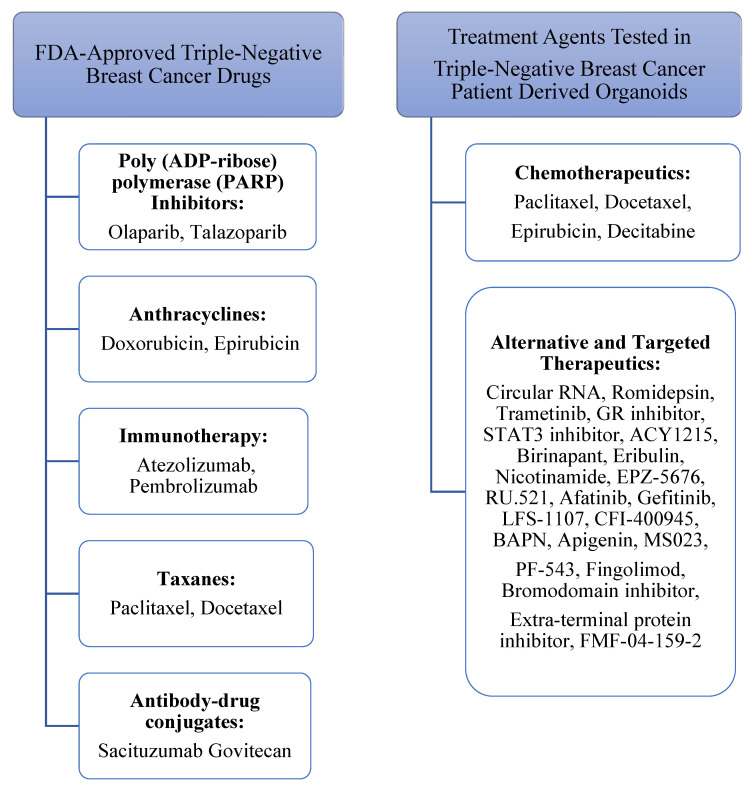
Current FDA-approved treatment agents for TNBC versus the drugs tested in and specific to TNBC patient-derived organoids.

**Table 1 biomedicines-11-00773-t001:** The use of organoids for chemotherapy sensitivity testing in TNBC.

Therapeutic Agents	Dose Ranges/IC50 Values	Main Results	References
Circular RNA	1.17–8.24 μM	Doxorubicin sensitivity increase	[35]
Decitabine	100 nM	DNMT protein level correlates with decitabine efficacy	[36]
Docetaxel, Epirubicin	Docetaxel dose: 1 nM Epirubicin logIC50 values = 0.3728 μM, 26.7300 μM	Decreased TNBC organoids’ viability Consistency of clinical response and response of tissue-derived organoids	[29,31]
Multiple Agents	20 μM–27.4 nM	High patient-derived organoid pharmaco-phenotyping association with clinical outcomes and previous treatment responses Patient-specific sensitivity predication for personalized therapy by patient-derived organoid pharmaco-phenotyping	[30]
MuV-U-Japan	Multiplicity of infection = 0–10	Potent killing effect in chemotherapy-resistant TNBC patient-derived xenograft organoids	[37]
Paclitaxel, Romidepsin, Trametinib	IC50 values = 10,500, 18,000 +/− 12.6, 6.5–24.9 μM	Organoid growth inhibition High IC_50_ paclitaxel value in TNBC patient-derived xenograft organoids CD44 and E-cadherin downregulation Enhanced apoptosis induction after SYTL4 knockdown	[32,33,34]

**Table 2 biomedicines-11-00773-t002:** The use of organoids for alternative and targeted therapy sensitivity testing in TNBC.

Therapeutic Agents	Dose Ranges/IC50 Values	Main Results	References
ACY1215	6.7 μM	Tektin4-deficient organoids favor the efficacy of HDAC6 inhibitors	[40]
Apigenin	1–50 μM	Organoid growth reduction	[49]
BAPN	25 mM	BAPN-Doxorubicin synergy leads to organoid shrinkage	[48]
Birinapant		50% of the employed TNBC patient-derived xenograft organoid lines showed remarkable sensitivity to birinapant	[41]
Bromodomain inhibitor, Extra-terminal protein inhibitor		Correlation with MARCO-TST expression	[52]
CFI-400945	1–50 nM	Synergy of radiation therapy and CFI-400945	[47]
EPZ-5676	10 nmol/L	Organoid growth reduction	[43]
Eribulin		Patient-derived xenograft organoid pharmaco-phenotyping predicates patient-specific sensitivities for personalized therapy	[41]
FMF-04-159-2		Organoid growth reduction, Cell proliferation inhibition, Apoptosis induction	[53]
GR inhibitor, STAT3 inhibitor	0.1–8 μM, 1–11 μM	Significant, synergistic, dose-dependent decrease in TNBC cell growth	[39]
LFS-1107	189.8 nM–1.187 μM	Organoid growth inhibition	[46]
MS023	0–10 μM	Correlation with basal interferon gene expression Upregulation of T1-helper cell chemokines and antigen presentation genes	[50]
Nicotinamide	IC50 values = 12–30 mM	Organoid growth reduction	[42]
PF-543, Fingolimod	10 μM, 1 μM	Higher efficacy in LAR positive TNBC	[51]
RU.521, Afatinib, Gefitinib	20 μM, 10 μM, 10 μM	Synergistic inhibitory effects on organoid growth	[45]

## Data Availability

Not applicable.

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
