# Peer review of "The Role of Patient-Derived Organoids in Triple-Negative Breast Cancer Drug Screening"

_biomedicines, 2023, doi:10.3390/biomedicines11030773_

Round 1
Reviewer 1 Report
Overall, very well-written manuscript that is well organized. A much-needed review article for this topic.
Major Concerns:
1. Generation of breast (cancer) organoids. While the authors describe well the protocol used by Sachs et al., an additional paragraph would be useful describing the protocol used by other authors subsequently. Were there any changes made to the protocol, and were there any particular ingredients that were thought to be critical or not useful? Were there any ingredients that may be particularly important for TNBC, since Dekkers et al. showed low efficiency of patient-derived organoid establishment.
2. An additional paragraph/section is necessary to explain what approaches the authors used to describe fidelity of the patient-derived organoids (PDOs), i.e. how similar are the PDOs to patient tumors? What genomic approaches were used? For the derivation of breast cancer organoids, what approaches were used to verify that the PDOs were derived from epithelial cells?
3. An additional paragraph/section is necessary to better describe how treatment response is compared in PDOs versus patient tumors. If IC50 values are used, how were they derived (e.g. single-cell suspension?) How do the PDO treatment response metrics compare to patient treatment response?
Minor Concerns
1. Page 1, Line 4. Sentence could perhaps be re-written as “cells do not overexpress the human epidermal growth factor receptor 2 (HER2) or receptors for estrogen and progesterone”.
2. Page 2, Last paragraph. It is not clear why 33 records were excluded after abstract review, and 11 studies excluded after text evaluation. Please explain.
3. Page 3. Please explain how ROCK inhibitor ameliorates culture conditions.
4. Page 4, Section 3, Line 6. Should read, “that characterize the more common hormone-receptor positive breast cancer entities”
5. Page 5. Section 4, Line 11. Is the word “predicated” or “predicted”?
Author Response
Overall, very well-written manuscript that is well organized. A much-needed review article for this topic.
Major Concerns:
- Generation of breast (cancer) organoids.While the authors describe well the protocol used by Sachs et al., an additional paragraph would be useful describing the protocol used by other authors subsequently. Were there any changes made to the protocol, and were there any particular ingredients that were thought to be critical or not useful? Were there any ingredients that may be particularly important for TNBC, since Dekkers et al. showed low efficiency of patient-derived organoid establishment.
Thank you for this useful suggestion. We have now added the work of Mohan et al. that comprehensively discusses significant mammary organoid studies with associated protocol details. Moreover, we have elaborated on the particular ingredients that seem to be critical or particularly important.
- An additional paragraph/section is necessary to explain what approaches the authors used to describe fidelity of the patient-derived organoids (PDOs), i.e. how similar are the PDOs to patient tumors?What genomic approaches were used? For the derivation of breast cancer organoids, what approaches were used to verify that the PDOs were derived from epithelial cells?
Following the reviewer’s suggestion, we have now added the approaches the authors used to describe fidelity of the patient-derived organoids.
- An additional paragraph/section is necessary to better describe how treatment response is compared in PDOs versus patient tumors.If IC50 values are used, how were they derived (e.g. single-cell suspension?) How do the PDO treatment response metrics compare to patient treatment response?
Thank you for this comment. Following also Reviewer’s 3 relevant suggestion, we have now added the IC50 values. Studies comparing breast organoids drug screening results with clinical treatment response may be found in the manuscript. In Section 4 for instance, we present the relevant works of Chen et al. and Shu et al.: ‘‘Moreover, Chen et al. derived organoids from patients with TNBC, drug-resistant, or metastatic tumors. Patient-derived organoid pharmaco-phenotyping was found to highly correlate with clinical outcomes and reflect previous treatment responses of the corresponding patients. Importantly, patient-derived organoids may predicate special sensitive drugs for personalized therapy, given that all treatments including at least one drug predicated to be sensitive by patient-derived organoids, achieved partial response, stable disease, or long disease-free survival (DFS), in TNBC patients. Similarly, Shu et al. matched the organoid drug sensitivity data with the patient’s clinical results, and outlined the consistency of the clinical response of TNBC patients with the response of tissue-derived organoids to neoadjuvant chemotherapy (docetaxel, epirubicin).’’.
Minor Concerns
- Page 1, Line 4. Sentence could perhaps be re-written as “cells do not overexpress the human epidermal growth factor receptor 2 (HER2) or receptors for estrogen and progesterone”.
We have now corrected the sentence accordingly.
- Page 2, Last paragraph. It is not clear why 33 records were excluded after abstract review, and 11 studies excluded after text evaluation.Please explain.
33 records were excluded after abstract review due to topic irrelevance. 11 studies were excluded after text evaluation, as they did not meet the inclusion criteria: Solely original research articles and scientific abstracts written in the English language, that explicitly reported on the development of three‐dimensional TNBC patient-derived organoid models.
- Page 3. Please explain how ROCK inhibitor ameliorates culture conditions.
We have now explained how ROCK inhibitor ameliorates culture conditions.
- Page 4, Section 3, Line 6. Should read, “that characterize the more common hormone-receptor positive breast cancer entities”
We have now corrected the sentence.
- Page 5. Section 4, Line 11.Is the word “predicated” or “predicted”?
We think that the word ’’predicated’’ is more appropriate in the given context.
Reviewer 2 Report
The authors provide essential information regarding recent advances in TNBC organoids in a developing field. Overall it is a nice summary of the findings. The authors could provide more details in figure 1 about specific search terms used and the URL of the databases. It will also be beneficial to figure out how to culture the organoids or provide any specific reference for the protocol. In figure 2: could the authors specify whether the drugs discovered by organoids were specific to organoids? It is unclear what benefit organoids provide that an otherwise 2d culture model would not have. The benefits of using organoids in drug discovery should be mentioned.
Author Response
The authors provide essential information regarding recent advances in TNBC organoids in a developing field. Overall it is a nice summary of the findings.
The authors could provide more details in figure 1 about specific search terms used and the URL of the databases.
Thank you for this useful suggestion. We have now added the specific search terms used and the URL of the databases in Figure 1.
It will also be beneficial to figure out how to culture the organoids or provide any specific reference for the protocol.
Thank you for this comment. We have now provided specific references [17] for the different protocols of breast (cancer) organoid generation in section 2.
In figure 2: could the authors specify whether the drugs discovered by organoids were specific to organoids? It is unclear what benefit organoids provide that an otherwise 2d culture model would not have. The benefits of using organoids in drug discovery should be mentioned.
The drugs discovered by organoids were specific to organoids. In the conclusion, we have now outlined the benefits of using organoids in drug discovery, especially when compared to 2d culture models.
Reviewer 3 Report
Manuscript ID: biomedicines-2227753
Title: The Role of Patient-Derived Organoids in Triple-Negative Breast Cancer Drug Screening
Authors Iason Psilopatis * , Amalia Mantzari , Kleio Vrettou , Stamatios Theocharis
Section: Cell Biology and Pathology
Special Issue: Organoids in Biomedical Research
The topic discussed by the authors in this review article is worthy of interest, and relevant in this field. But several points need to be addressed to improve the quality of the manuscript.
1. Introduction
Why did the authors focus their research only on two databases (MEDLINE and LIVIVO)? Enlarging the explored databases may give more relevant information on the topic of interest.
2. Generation of breast (cancer) organoids
The culture conditions of breast organoids culture are not well discussed. The authors mainly focus on factors that may influence organoids culture without giving minimal requirements that come out form different articles analysed. The cultures conditions used in different articles should be discussed, and perspectives should be given toward breast organoids culture optimization. The impression that we get from this review is that there is no consensual method for the generation of breast cancer organoid. The authors should come out with similarities and differences observed in methods used for the generation of breast cancer organoids in the selected articles.
Tables: Presenting main results in a form of text in tables is not appropriate, it is better to include measurable parameters used to evaluate the chemotherapy sensitivity.
6. Conclusions
Studies comparing breast organoids drug screening results with clinical treatment response are not found in the manuscript. This should be discussed to show the importance of this approach for personalized breast cancer treatment.
Author Response
Manuscript ID: biomedicines-2227753
Title: The Role of Patient-Derived Organoids in Triple-Negative Breast Cancer Drug Screening
Authors Iason Psilopatis *, Amalia Mantzari, Kleio Vrettou, Stamatios Theocharis
Section: Cell Biology and Pathology
Special Issue: Organoids in Biomedical Research
The topic discussed by the authors in this review article is worthy of interest, and relevant in this field. But several points need to be addressed to improve the quality of the manuscript.
- Introduction
Why did the authors focus their research only on two databases (MEDLINE and LIVIVO)? Enlarging the explored databases may give more relevant information on the topic of interest.
MEDLINE and LIVIVO represent two well-established and extensive databases that incorporate most relevant articles on the topic of focus and, hence, lay the basis for every up-to-date (systematic) review article. By searching both databases, we were able to identify 25 relevant articles which, to our knowledge, constitute the current, most comprehensive review of the literature on the role of organoids in TNBC drug screening.
- Generation of breast (cancer) organoids
The culture conditions of breast organoids culture are not well discussed. The authors mainly focus on factors that may influence organoids culture without giving minimal requirements that come out form different articles analysed. The cultures conditions used in different articles should be discussed, and perspectives should be given toward breast organoids culture optimization. The impression that we get from this review is that there is no consensual method for the generation of breast cancer organoid. The authors should come out with similarities and differences observed in methods used for the generation of breast cancer organoids in the selected articles.
Thank you for this useful suggestion. We now discuss the culture conditions of breast organoid culture and refer to the similarities and differences observed in different methodologies.
Tables: Presenting main results in a form of text in tables is not appropriate, it is better to include measurable parameters used to evaluate the chemotherapy sensitivity.
Following the reviewer’s suggestion, we have now added the IC50 values as measurable evaluation parameters in the context of chemotherapy sensitivity.
- Conclusions
Studies comparing breast organoids drug screening results with clinical treatment response are not found in the manuscript. This should be discussed to show the importance of this approach for personalized breast cancer treatment.
Studies comparing breast organoids drug screening results with clinical treatment response may be found in the manuscript. In Section 4 for instance, we present the relevant works of Chen et al. and Shu et al.: ‘‘Moreover, Chen et al. derived organoids from patients with TNBC, drug-resistant, or metastatic tumors. Patient-derived organoid pharmaco-phenotyping was found to highly correlate with clinical outcomes and reflect previous treatment responses of the corresponding patients. Importantly, patient-derived organoids may predicate special sensitive drugs for personalized therapy, given that all treatments including at least one drug predicated to be sensitive by patient-derived organoids, achieved partial response, stable disease, or long disease-free survival (DFS), in TNBC patients. Similarly, Shu et al. matched the organoid drug sensitivity data with the patient’s clinical results, and outlined the consistency of the clinical response of TNBC patients with the response of tissue-derived organoids to neoadjuvant chemotherapy (docetaxel, epirubicin).’’.
Round 2
Reviewer 1 Report
Good work.
Author Response
Thank you.
Reviewer 3 Report
The authors have addressed most of the issues, but I have a suggestion regarding different tables of the manuscript. It will suggest that a column (Dose range or IC50 values of therapeutic agents) should be included in each table in order to make it uniform in case the authors cannot get the IC50 values of all the therapeutic agents.
Author Response
We have now included a column with dose ranges or IC50 values.